# Immunogenomic Identification for Predicting the Prognosis of Cervical Cancer Patients

**DOI:** 10.3390/ijms22052442

**Published:** 2021-02-28

**Authors:** Qun Wang, Aurelia Vattai, Theresa Vilsmaier, Till Kaltofen, Alexander Steger, Doris Mayr, Sven Mahner, Udo Jeschke, Helene Hildegard Heidegger

**Affiliations:** 1Department of Obstetrics and Gynecology, University Hospital, LMU Munich, 80377 Munich, Germany; wqyxdz888@163.com (Q.W.); aurelia.vattai@med.uni-muenchen.de (A.V.); Theresa.Vilsmaier@med.uni-muenchen.de (T.V.); till.kaltofen@med.uni-muenchen.de (T.K.); sven.mahner@med.uni-muenchen.de (S.M.); Helene.heidegger@med.uni-muenchen.de (H.H.H.); 2Klinik für Innere Medizin I, Technische Universität München, 80333 Munich, Germany; alexander.steger@tum.de; 3Department of Pathology, LMU Munich, 80377 Munich, Germany; doris.mayr@med.uni-muenchen.de; 4Department of Obstetrics and Gynaecology, University Hospital Augsburg, Stenglinstr. 2, 86156 Augsburg, Germany

**Keywords:** cervical cancer, tumor immune, bioinformatics analysis, TCGA, KEGG

## Abstract

Cervical cancer is primarily caused by the infection of high-risk human papillomavirus (hrHPV). Moreover, tumor immune microenvironment plays a significant role in the tumorigenesis of cervical cancer. Therefore, it is necessary to comprehensively identify predictive biomarkers from immunogenomics associated with cervical cancer prognosis. The Cancer Genome Atlas (TCGA) public database has stored abundant sequencing or microarray data, and clinical data, offering a feasible and reliable approach for this study. In the present study, gene profile and clinical data were downloaded from TCGA, and the Immunology Database and Analysis Portal (ImmPort) database. Wilcoxon-test was used to compare the difference in gene expression. Univariate analysis was adopted to identify immune-related genes (IRGs) and transcription factors (TFs) correlated with survival. A prognostic prediction model was established by multivariate cox analysis. The regulatory network was constructed and visualized by correlation analysis and Cytoscape, respectively. Gene functional enrichment analysis was performed by Gene Ontology (GO) and Kyoto Encyclopedia of Genes and Genomes (KEGG). A total of 204 differentially expressed IRGs were identified, and 22 of them were significantly associated with the survival of cervical cancer. These 22 IRGs were actively involved in the JAK-STAT pathway. A prognostic model based on 10 IRGs (*APOD*, *TFRC*, *GRN*, *CSK*, *HDAC1*, *NFATC4*, *BMP6*, *IL17RD*, *IL3RA*, and *LEPR*) performed moderately and steadily in squamous cell carcinoma (SCC) patients with FIGO stage I, regardless of the age and grade. Taken together, a risk score model consisting of 10 novel genes capable of predicting survival in SCC patients was identified. Moreover, the regulatory network of IRGs associated with survival (SIRGs) and their TFs provided potential molecular targets.

## 1. Introduction

Cervical cancer is primarily caused by the infection of high-risk human papillomavirus (hrHPV), and it ranks the fourth most common cancer in females globally [1]. There are approximately 527,000 new cases of cervical cancer and 265,000 related deaths annually [2]. Among all pathological types of cervical cancer, squamous cell carcinoma (SCC) and adenocarcinoma account for 80–85% and 15–20%, respectively [3]. Presently, although the development of surgery, radiation therapy, and chemotherapy, the rates of recurrence and metastasis in patients with late-stage cervical cancer are still up to 40.3% and 31%, respectively [4]. The prognosis of patients with metastatic cervical cancer remains poor, with a median survival of 8–13 months [5]. Therefore, it is urgently necessary to identify effective predictive biomarkers and molecular mechanisms involved in the prognosis of cervical cancer, which may help find better predictive and therapeutic targets.

Increasing attention has been paid to research on immunotherapy for cervical cancer. Zhou J et al. have shown that IFNα-expressing amniotic fluid-derived mesenchymal stem cells can suppress HeLa cell-derived tumors in a mouse model [6]. An HPV vaccine containing some HPV16 E7 peptides presents an anti-tumor effect in tumor-free mice [7]. Jung KH et al. have enhanced the efficacy of T cells by optimizing the maturation and function of dendritic cells with lipopolysaccharide (LPS) and interferon (IFN)γ, adding interleukin (IL)-21 during priming, and depleting memory T cells, and the reliable expansion of T cells specific for oncoproteins E6 and E7 has been achieved [8]. Currently, the treatment for immune checkpoints, such as cytotoxic T lymphocyte 4 (CTLA-4), programmed death protein 1 (PD-1), and its ligand (PD-L1), has shown initial success against cervical cancer [9]. Since the great potential in immunotherapy, more molecular mechanisms need to be explored to improve the immunotherapy. Transcriptome profiling, as a high-throughput research approach, has been applied to many cancer studies. For example, i n non-small cell lung cancer and thyroid cancer, the prognostic value of immune-related genes (IRGs) has been analyzed with the data acquired from sequencing [10,11]. However, the clinicopathological correlation and prognostic significance of IRGs in cervical cancer remain largely undetermined.

In the present study, we identified 204 differentially expressed IRGs between cervical tumor and para-tumor tissues, and an individualized prognostic prediction model based on risk scores was constructed for patients with early-stage SCC. Functional enrichment analysis showed that the IRGs associated with survival (SIRGs) were mainly involved in the receptor-ligand activity and JAK-STAT signaling pathway. The receiver operating characteristic (ROC) curve and risk curve verified the moderate efficacy of the predictive model. The area under the curve (AUC) was 0.738. The overall survival (OS) in the high-risk group was significantly lower, compared with the low-risk group (*p* = 2.702 × 10^5^). Moreover, Kaplan-Meier analysis showed that the model worked steadily, regardless of age and grade. The risk curve showed that the higher the risk score, the more the deaths, and the shorter the OS. Besides, according to univariate and multivariate cox regression analyses, the risk score could be an independent risk factor (HR = 3.170, 95% CI [1.701–5.910], *p* = 0.0001) adjusted by age, grade, stage, and histological type. Interactive transcription factors (TFs) for IRGs associated with survival were also explored. These results could offer not only a promising prediction model for cervical cancer prognosis but also molecular targets to study the immunity mechanism for cervical cancer progression. 

## 2. Results

### 2.1. Identification of Differentially Expressed Genes (DEGs)

A total of 2240 DEGs, including 1412 down-regulated and 1928 up-regulated ones (Appendix A and Figure 1a), were identified by comparing the gene expression data between three para-tumor tissue specimens and 286 primary cervical tumor tissue specimens. By comparing the IRGs obtained from the Immunology Database and Analysis Portal (ImmPort) database [12] with the DEGs, 204 IRGs overlapped with DEGs were selected, including 115 down-regulated, and 89 up-regulated ones (Appendix A and Figure 1b). Gene Ontology (Go) analysis revealed that these differentially expressed IRGs were significantly associated with tumor-related biological processes. Moreover, “epithelial cell proliferation”, “receptor complex” and “growth factor binding” were the most frequent biological terms in biological processes, cellular components, and molecular functions, respectively (Table 1). Cytokine-cytokine receptor interaction was the most frequently identified function of potential pathways by Kyoto Encyclopedia of Genes and Genomes (KEGG) (Figure 1c).

### 2.2. Identification of Differentially Expressed SIRGs

Since survival time and status are important for the prognostic evaluation, it seems to be feasible to evaluate the prognosis of patients based on the expressions of genes associated with survival. First, we identified 22 differentially expressed SIRGs among 212 cervical cancer cases with complete OS data. GO analysis revealed that tumor-related biological process was the most frequently implicated term (Table 2). JAK-STAT signaling pathway was the most frequently identified pathway analyzed by KEGG (Figure 2a). A forest plot of hazard ratios indicated that 10 genes were significant protective factors, and 12 genes were significant adverse factors (Figure 2b). Protein-protein interaction (PPI) network analysis demonstrated that TYK2, CSK, PTPN6, and IL3RA were the hub genes, which were screened out based on the criteria of correlation coefficient 0.3 and the number of interactive genes no less than 3 (Figure 2c). These hub genes were actively involved in the JAK-STAT signaling pathway (Figure 2d). 

### 2.3. Construction of the Prognostic Model

A total of 10 IIRGs were screened out from the 22 SIRGs using the multivariate COX analysis, which were independent factors for the survival status and time of 212 cervical cancer cases. Table 3 shows their correlation coefficients. A prognostic model was established as follows:

Risk score = [Expression level of APOD × (−0.06584)] + [Expression level of TFRC × 0.004018] + [Expression level of GRN × 0.00648)] + [Expression level of CSK × (−0.04999)] + [Expression level of HDAC1 × (−0.01997)] + [Expression level of NFATC4 × 0.129489 + [Expression level of BMP6 × 0.055054] + [Expression level of IL17RD × 0.124096]+ [Expression level of IL3RA × (−0.22745)] + [Expression level of LEPR × 0.520483]. 

### 2.4. Efficacy Verification of the Prognostic Model

To verify the efficacy of the prognostic model, we performed a Kaplan-Meier survival analysis for the 212 cases. The OS rate of the high-risk group was significantly lower compared with the low-risk group (*p* < 0.001). The result showed that the prognostic model could distinguish different clinical outcomes from cervical cancer patients (Figure 3a). Besides, we also performed an ROC analysis (Figure 3b). The AUC was 0.738, suggesting moderate accuracy for the prognosis in cervical cancer. Moreover, the model still worked in subgroup analyses of age (Appendix A) and grade (Appendix A), while it could only work in subgroup analyses of SCC (Appendix A) and FIGO I stage (Appendix A) subgroups, suggesting that the model was steady in patients with early-stage SCC, regardless of age or grade. The risk curve showed that patients could be divided into the high-risk group and low-risk group according to the median risk score, and the survival time of the high-risk group was lower compared with the low-risk group. Moreover, the number of death was greater in the high-risk group compared with the low-risk group (Figure 3c,d). The DEGs involved in the prognostic model were shown in the heat map, which was consistent with the trend shown by the correlation coefficients (Figure 3e, Table 3). 

Univariate analysis and multivariate analysis indicated that the risk score could be an independent predictor (HR = 3.170, 95% CI [1.701–5.910], *p* = 0.001) after other clinicopathological characteristics, such as age, grade, and Figo stage, were adjusted (Table 4). 

Therefore, the predictive model could be a reliable and steady method to judge the clinical outcomes for SCC patients with FIGO I stage. For example, the 5-year survival rate of the low-risk group and high-risk group were about 80% and 55%, respectively.

### 2.5. The Clinical Significance of IIRGs

We analyzed the differences of IIRGs in clinicopathological characteristics, including age, grade, FIGO stages, of the 212 cases to determine the relationship between the risk score and clinical parameters. The expressions of IL17RD and TFRC were significantly higher in the group of patients aged over 45 years (Figure 4a,b). The expression of HDCA1 was significantly lower in the group of grades 3&4 compared with the group of grades 1&2 (Figure 4c). 

### 2.6. TFs Regulatory Network

To explore the potential molecular mechanisms of SIRGs, we investigated the TFs associated with 22 SIRGs. We selected differentially expressed 74 TFs by intersecting the list of TFs with DEGs (Figure 5a). Among these 74 TFs, five TFs were significantly associated with survival (Figure 5b). Based on the criteria of correlation coefficient = 0.3, and *p* = 0.001, a regulatory network was constructed using these five TFs and SIRGs (Figure 5c). 

## 3. Discussion

Cervical cancer is caused by the persistent infection of hrHPV [13,14,15]. Although surgery, chemoradiotherapy, anti-angiogenic medicine, and even the new immunotherapy have been applied for cervical cancer treatment, the prognosis remains poor at the late stage [4,9,16]. Therefore, research on effective prognostic biomarkers and new molecular mechanisms has drawn increasing attention. Kidd EA et al. have found that the standardized uptake value for F-18 fluorodeoxyglucose is a sensitive predictive biomarker for the survival of cervical cancer patients [17]. Luo W et al. have identified a 6-lncRNA signature, which can be regarded as novel diagnostic biomarkers for cervical cancer [18]. Li X et al. have identified a histone family gene signature for predicting the prognosis of cervical cancer patients [19]. These studies have provided an elemental knowledge of the pathogenesis of cervical cancer at the genetic level. However, the prognostic role of immunogenomics in cervical cancer remains largely undetermined. In the present study, we performed a comprehensive analysis of IRGs in cervical cancer, which might enhance our knowledge of their clinical value and help us understand potential molecular mechanisms. Moreover, these IRGs might act as valuable clinical biomarkers or therapeutic targets. Besides, we constructed a prognostic model that could help assess potential clinical outcomes of cervical cancer patients. 

Tumor immune microenvironment can promote the progression of cervical cancer, including cancer cell proliferation, invasion, metastasis, immunosuppression and tissue remodeling, fibrosis, and angiogenesis. For example, CXCL12 induces mononuclear phagocytes to release HB-EGF, triggering anti-apoptotic and proliferative signals in Hela cells [20]. D-dopachrome tautomerase (D-DT), a homolog of macrophage migration inhibitory factor (MIF), can promote the invasion of cervical cancer cells when it is overexpressed [21]. An altered balance in IL-12p70 and IL-10 production can weaken T cell proliferation in cervical cancer [22]. These studies suggest the importance of immunity in cervical cancer progression. Therefore, it is necessary to identify differentially expressed IRGs. Genome profile alterations cause tumorigenesis. We identified alternations in immunogenomic profiles to study the effect of alternations on the immune microenvironment and clinical prognosis. Gene functional enrichment analysis suggested that these genes were mainly involved in growth factor (GF) activity. GFs actively act in the pathogenesis of cervical cancer. Notably, these GFs are correlated to proliferation, aggression, and migration [23,24,25]. Therefore, these GFs could also be used to monitor metastasis, assess survival, and identify potential drug targets as clinical biomarkers. 

Among the 20 SIRGs, no related reports have explored the roles of DUOX1, GRN, CSK, CD79, NFATC4, EPGN, TGFA, IL17RD, LEPR, N2RF1, and TRAV26-1 in cervical cancer. APOD is down-regulated in cervical cancer compared with normal cervix [26]. TFRC expression is up-regulated in cervical cancer compared with normal cervix [27]. A previous study has found that there is a significant correlation between cervical cancer and the polymorphism of rs1041981 in the LTA gene [28]. The rare allele (A) of SNP rs2239704 in the 5′ UTR of the LTA gene is significantly associated with increased risks of cervical cancer [29]. F2RL1 is overexpressed in cervical cancer cell lines and significantly correlated with poor OS [30,31]. HGF overexpression in lesions of cervical cancer has been reported to be related to a poor prognosis [32]. HDAC1/DNMT3A-containing complex is associated with the suppression of cancer stem cells in cervical cancer [33]. HGF can induce migration and invasion of cervical cancer cells [34].] BMP6 may participate in invasion and metastasis in cervical cancer [35]. The proportion of CD123(+) dendritic cells is significantly lower in the peripheral blood of cervical cancer patients compared with the controls [36]. A higher frequency of Nrp1(+) T-regs frequency suppresses the immune response against distant cervical cancer cells [37]. The above-mentioned results are consistent with our current findings. However, Tyk2 is confirmed to be overexpressed in SCC [38], which is different from our study. Low-throughput experiments, such as Western blotting analysis, are required to verify the factual expression. PTPN6 is positively correlated with HPV infection in cervical cancer with the explanation of cell defense reaction [39]. 

To explore molecular mechanisms underlying the potential clinical importance, we constructed a TF-mediated network that could regulate hub IRGs. Among the SDETFs, Foxp3 is significantly associated with FIGO stage and tumor size [40]. Foxp3 is associated with lymphangiogenesis of cervical cancer [41]. FoxP3 has been confirmed to be highly expressed in cervical cancer, and it facilitates the proliferation and invasiveness and inhibits the apoptosis of cervical cancer cells [42]. In conclusion, Foxp3 is a risk factor for the survival of cervical cancer, which is consistent with our current findings. CBX7 inhibits the proliferation of cervical cancer cells [43]. LTA inhibits the proliferation of CD4(+) T-cells in a FoxP3(+) Treg-dependent manner in patients with chronic hepatitis C, suggesting that LTA acts on FoxP3 [44]. Therefore, previous studies provide limited information about the mechanisms of 10 IRGs in the survival of cervical cancer.

The effects of the JAK/STAT pathway and the persistent activation of STAT3 and STAT5 during the process of tumor cell proliferation, cycling, and invasion have made it a favorite treatment target. In cervical cancer, the activated JAK/STAT signaling pathway by Bcl-2 promotes cell viability, migration, and invasion [45]. There is a strong association between HPV infection and STAT-3 overexpression in cervical cancer [46]. The expression of STAT3 has been proposed as a poor prognostic factor in cervical cancer [47]. STAT5 protein is up-regulated and associated with the severity of cervical cancer [48]. Moreover, overexpression of STAT-5 elevates the STAT-3 expression compared with the normal controls [46]. Therefore, JAK/STAT signaling may play an important role in cervical carcinogenesis. Moreover, up-regulation of PAR2 (F2RL1) induces the proliferation of cervical cancer cells by activating STAT3 [31], which is consistent with our current results that overexpression of F2RL1 was involved in the JAK/STAT signaling pathway. 

In the present study, we created an immune-based prognostic signature to monitor the immune status and assess the prognosis for cervical cancer patients. Previously, Wu HY et al. (2020) have constructed a prognostic index based on percent-splice-in values in SCC [49]. Eun Jung Kwon et al. (2020) have explored genomic alterations and developed a risk index model that can monitor HPV- related bladder cancer [50]. Cai LY has created a risk score model based on differentially expressed glycolysis-related genes, and the model can predict the prognosis of cervical cancer patients [51]. Recently, Zhao S et al. (2020) have constructed a 4-gene prognostic risk score model in CESC by identifying DEGs [52]. Beyond the above-mentioned studies, there are also many studies about the prognostic model [53,54]. Compared with the previous studies, our prognostic model could assecss immune-genomic profiles. Moreover, we constructed a TF-mediated regulatory network, which provided a more detailed mechanism of IIRGs. Our prognostic index, based on 10 differentially expressed IIRGs in cervical cancer, demonstrated favorable clinical viability. Our data showed that the risk score model performed moderately and steadily in prognostic predictions in patients with early-stage cervical cancer. 

We must point out that only three control specimens were acquired in the present study. Although it met the minimum requirements for the biological repeat, insufficient control samples tended to cause larger errors. Therefore, more other experiments are still necessary to validate the transcriptome results.

## 4. Materials and Methods 

### 4.1. Gene Expression Data and Clinical Data Collection

FPKM transcriptome RNA-sequencing data and clinical data of cervical samples were downloaded from TCGA data portal (https://portal.gdc.cancer.gov/) on 1 December 2020. FPKM transcriptome RNA-sequencing data were derived from 289 samples, including three para-tumor tissue specimens (Table 5) and 286 primary cervical tumor tissues. As to para-tumor slides, the percentage of tumor cells, lymphocytes, necrosis, infiltrated monocytes, infiltrated neutrophil was 0, and the percentage of normal cells was 100%. The percentage of stroma cells of TCGA-FU-A3EO was 0, while that of the other two para-tumor tissue specimens was missing. As to tumor slides, the percentage of tumor cells was 80% (70%, 90%), the percentage of infiltrated lymphocytes was 7.5% (2%, 40%), the percentage of infiltrated monocytes was 0% (0%, 10%), the percentage of necrosis was 2% (0, 5%), the percentage of infiltrated neutrophil was 1% (0%, 20%), and the percentage of normal cells was 0% (0%, 5%). The para-tumor slides were collected from Christiana Healthcare, International Genomics Consortium, and Montefiore Medical Center, respectively. The samples were obtained from 253 patients with SCC and 31 patients with cervical adenocarcinoma. As to the clinical data, there were 212 cases with clinical data including age, grade, FIGO stage, survival status, and OS no less than 90 days. The clinical characteristics of the 212 cases were shown in Table 6. 

The list of IRGs was downloaded from the ImmPort database, which is a powerful public database with hundreds of downloads per month [12]. 

### 4.2. Analysis of DEGs

To identify the DEGs between three para-tumor and 289 cervical tumor tissue specimens, an R language script was made, and a limma package (http://www.bioconductor.org/packages/release/bioc/html/limma.html) on 2 January 2020 was downloaded and performed by the R software. The mean value of each gene expressed in para-tumor samples and cervical tumor samples was calculated by the script, and repeated genes were deleted by limma package. Wilcoxon-test was applied to compare the difference in expression. The false discovery rate (FDR) <0.05 and log2 |fold change| > 1 were set as the cutoff values to identify the DEGs. Differentially expressed IRGs were extracted by intersecting the results of DEGs and the list of IRGs by using the R language. The DEGs were presented using the pheatmap package and volcano plot script by R software. 

The DEGs and differentially expressed IRGs were then subjected to clusterProfiler package in R for GO and KEGG pathway enrichment analysis to determine their potential functions and pathways. 

### 4.3. Screening of Differentially Expressed SIRGs 

IRGs significantly associated with survival of 212 tumor cases were identified from all the differentially expressed IRGs through univariate cox analysis (*p* < 0.05). These IRGs were defined as SIRGs. Forest map script was performed by R software.

The SIRGs were then subjected to cluster-Profiler package for GO and KEGG pathway enrichment analysis to determine their potential functions and pathways [13].

### 4.4. Construction of the Prognostic Model

R language script was made to perform multivariate cox analysis of SIRGs acquired by univariate cox analysis. Those IRGs in the model were defined as IIRGs. The risk score, as the indicator of the prognostic model, was calculated as the sum of the product of each IIRG expression and its regression coefficient. According to the median risk score of 1.276, all the 212 cases were divided into the high-risk group and low-risk group. 

### 4.5. Construction of the Regulatory Network of SIRGs and Their TFs

The list of TFs related to cancer was downloaded from the Cistrome database (http://cistrome.org/) on 3 January 2020. This public database has collected the set of cis-acting targets of a trans-acting factor on a genome-wide scale. The TFs were intersected with DEGs, and the differentially expressed TFs (DETFs) were obtained by R language. Moreover, univariate cox analysis was carried out to identify DETFs significantly associated with survival (SDETFs).

Spearman correlation analysis between SDETFs and SIRGs in tumor tissues from the same samples was performed by R software. The correlation coefficient >0.3 and *p* < 0.001 were set as the standard for screening the correlated SDETFs (CSDETFs) with SIRGs. Therefore, the CSDETFs were acquired. Cytoscape software version 3.7.1 was applied to display the regulation relationship between CSDETFs and SIRGs.

### 4.6. Validaion of the Risk Score Model

Survival analysis was performed between the high-risk group and low-risk group in terms of age (age ≤ 45, and age > 45), grade (1–2, and grade3–4), cancer type (cervical adenocarcinoma and SCC), and FIGO stage (I, II–III, IVa, and IVb) by SPSS. A *p* value of less than 0.05 was considered statistically significant.

### 4.7. Statistical Analysis

The screening analysis of DEGs was performed by the Wilcoxon-test. Correlation analysis between SDETFs and SIRGs was carried out by Spearman correlation. DEG expression in each clinicopathological characteristic was tested by theWilcoxon-test. AUC of the survival ROC curve was calculated by the ROC package in R software. A *p* value of less than 0.05 was considered statistically significant [14].

## 5. Conclusions

Collectively, we adopted the bioinformatic method to comprehensively analyze the differentially expressed IRGs in cervical cancer and identified 204 tumor-associated IRGs. Among them, we also identified 22 SIRGs and constructed an individual predictive model with moderate accuracy and stability for prognostic prediction in SCC patients with FIGO I. We further explored the KEGG pathway and regulatory network between survival-associated TFs and SIRGs. However, the exact mechanism underlying how these genes affected the prognosis of cervical cancer should be verified by more accurate experiments.

## Figures and Tables

**Figure 1 ijms-22-02442-f001:**
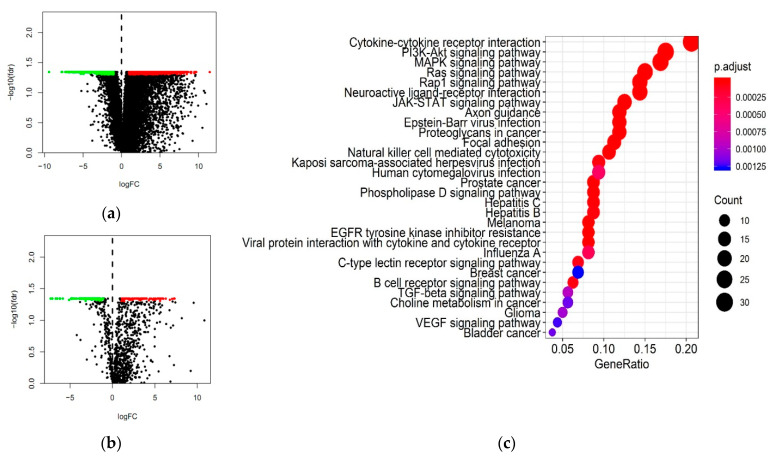
DEGs. (**a**) Volcano plot of DEGs between primary cervical cancer and para-tumor tissues. (**b**) Volcano plot of differentially expressed IRGs. Green dots, down—regulated genes; red dots, up-regulated genes, black dots, no DEGs. (**c**) The results of KEGG analysis. IRGs, Immune—related genes.

**Figure 2 ijms-22-02442-f002:**
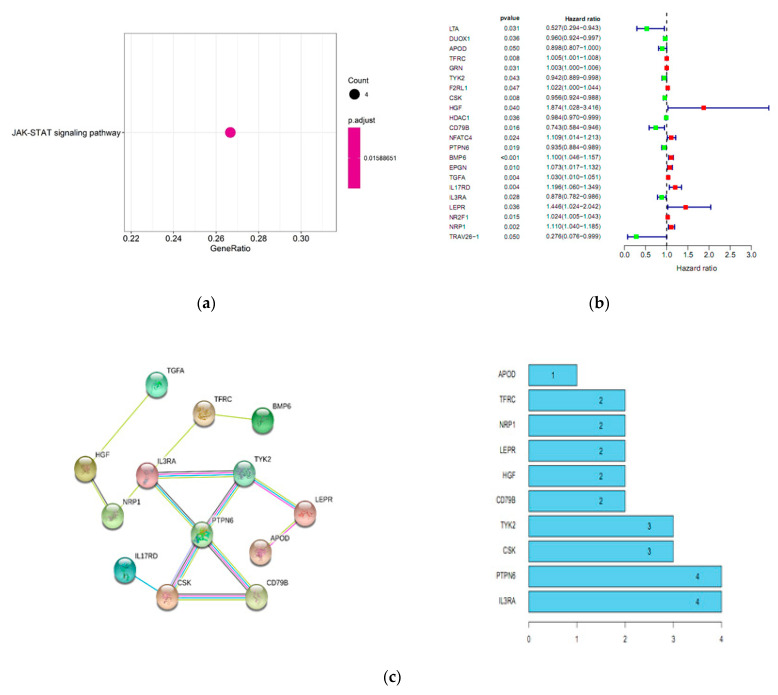
Identification of SIRGs. (**a**) The most significant KEGG pathways for SIRGs. (**b**) A forest plot of hazard ratios. The left is the list of SIRGs and their prognostic values showing as name, *p*-value, and the hazard ratio (95% CI), and the right is the relevant forest plot; Green bar, protective factor; red bar, adverse factor. (**c**) PPI network. The left is the PPI network, and the right is the number of interactive genes for each gene. (**d**) The most significant KEGG pathways for the hub SIRGs. SIRGs, immune-related genes associated with survival.

**Figure 3 ijms-22-02442-f003:**
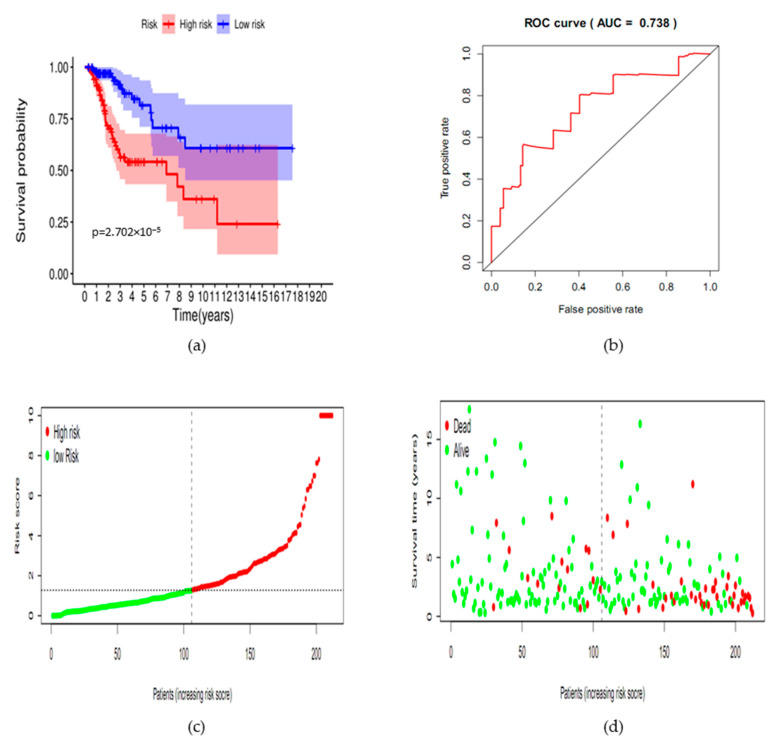
Verification of the efficacy of the prognostic model. (**a**) Kaplan-Meier plots demonstrated that the prognostic model could distinguish different clinical outcomes from cervical cancer patients (*p* < 0.05). Blue represents the low-risk group; red represents the high-risk group. (**b**) The ROC for verifying the accuracy of the predictive model and AUC for the risk score model displayed moderately accuracy in the cancer Genome Atlas (TCGA) dataset. (**c**) Value of risk score in cervical cancer patients. Both the horizontal axis and the vertical axis represent risk score. From left to right, the risk score is increasing; red dot represents the high- risk case; green dot represents the low-risk case; (**d**) survival status and time in the two risk groups. From left to right, the risk score is increasing. The vertical axis represents the survival time. (**e**) Heatmap of the differentially expressed SIRGs involved in the prognostic model. From left to right, the risk score is increasing. Blue represents a high-risk case. Red represents a low-risk case.

**Figure 4 ijms-22-02442-f004:**
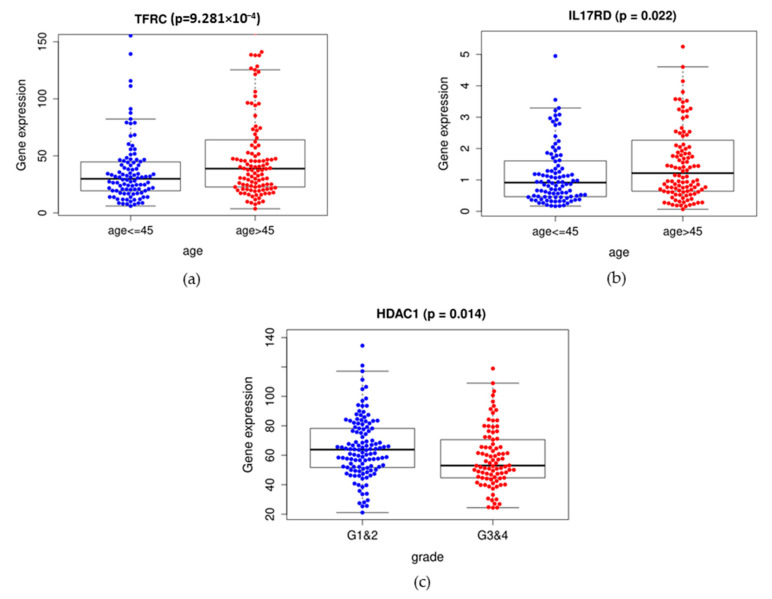
The clinical significance of IIRGs. The box plots showed that the expressions of TFRC, IL17RD, and HDAC1 were significantly different in subgroups of age and grade. (**a**,**b**) Blue represents the group of age ≤ 45years, and red represents the group of age > 45 years. (**c**) Blue represents the group of grades 1 and 2, and red represents the group of grades 3 and 4.

**Figure 5 ijms-22-02442-f005:**
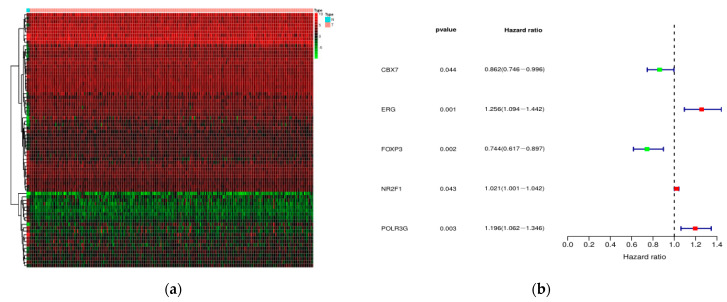
The construction of a regulatory network between SDETFs and SIRGs. (**a**) Heatmap of the SDETFs. (**b**) A forest plot of hazard ratios. The left is the list of SDETFs and their prognostic values showing as name, *p*-value, and the hazard ratio (95% CI), and the right is the relevant forest plot. Green bar, protective factor; red bar, adverse factor. (**c**) A regulatory network between SDETFs and SIRGs. Triangle, TFs; Roundness, SIRGs; Red roundness, the overexpressed SIRGs; Green roundness, down-expressed SIRGs. Red line, the TFs up-regulate SIRGs; Green line, the TFs down-regulate SIRGs.

**Table 1 ijms-22-02442-t001:** GO analysis.

ID	Description	*p*-Adjust	Count
GO:0050673	epithelial cell proliferation	1.78 × 10^−18^	36
GO:0050678	regulation of epithelial cell proliferation	1.78 × 10^−18^	34
GO:0050679	positive regulation of epithelial cell proliferation	2.34 × 10^−16^	25
GO:0018108	peptidyl-tyrosine phosphorylation	1.52 × 10^−15^	30
GO:0018212	peptidyl-tyrosine modification	1.55 × 10^−15^	30
GO:0032103	positive regulation of response to external stimulus	7.98 × 10^−14^	27
GO:0001667	ameboidal-type cell migration	1.22 × 10^−13^	31
GO:0010631	epithelial cell migration	1.26 × 10^−13^	28
GO:0090132	epithelium migration	1.39 × 10^−13^	28
GO:0090130	tissue migration	1.93 × 10^−13^	28
GO:0060326	cell chemotaxis	2.54 × 10^−12^	23
GO:0010632	regulation of epithelial cell migration	6.51 × 10^−12^	24
GO:0043410	positive regulation of MAPK cascade	9.99 × 10^−12^	30
GO:0050900	leukocyte migration	1.07 × 10^−11^	29
GO:0043235	receptor complex	1.08 × 10^−11^	23
GO:0009897	external side of plasma membrane	3.19 × 10^−7^	15
GO:0060205	cytoplasmic vesicle lumen	1.02 × 10^−6^	18
GO:0031983	vesicle lumen	1.02 × 10^−6^	18
GO:0034774	secretory granule lumen	2.17 × 10^−6^	17
GO:0031012	extracellular matrix	3.89 × 10^−6^	20
GO:0062023	collagen-containing extracellular matrix	6.87 × 10^−6^	18
GO:0098552	side of membrane	3.49 × 10^−5^	16
GO:0022624	proteasome accessory complex	9.42 × 10^−5^	5
GO:0005912	adherens junction	0.00042719	17
GO:0045121	membrane raft	0.000466985	13
GO:0098857	membrane microdomain	0.000466985	13
GO:0098589	membrane region	0.000619958	13
GO:0031093	platelet alpha granule lumen	0.001187147	6
GO:0005925	focal adhesion	0.001187147	14
GO:0019838	growth factor binding	4.90 × 10^−18^	22
GO:0048018	receptor ligand activity	3.61 × 10^−17^	33
GO:0030545	receptor regulator activity	1.93 × 10^−16^	33
GO:0019199	transmembrane receptor protein kinase activity	1.44 × 10^−15^	16
GO:0008083	growth factor activity	2.21 × 10^−14^	20
GO:0005126	cytokine receptor binding	1.90 × 10^−11^	20
GO:0019955	cytokine binding	2.16 × 10^−11^	15
GO:0005178	integrin binding	1.19 × 10^−10^	15
GO:0004713	protein tyrosine kinase activity	2.39 × 10^−10^	15
GO:0042562	hormone binding	2.39 × 10^−10^	13
GO:0050431	transforming growth factor beta binding	1.42 × 10^−9^	8
GO:0005539	glycosaminoglycan binding	3.49 × 10^−9^	17
GO:0004714	transmembrane receptor protein tyrosine kinase activity	4.41 × 10^−9^	10
GO:0005125	cytokine activity	6.15 × 10^−9^	15
GO:0003707	steroid hormone receptor activity	1.06 × 10^−8^	10

Note: Green bar, biological process. Blue bar, cellular components. Pink bar, molecular function.

**Table 2 ijms-22-02442-t002:** GO analysis.

ID	Description	*p* Adjust	Count
GO:0018108	peptidyl-tyrosine phosphorylation	9.23 × 10^−5^	7
GO:0018212	peptidyl-tyrosine modification	9.23 × 10^−5^	7
GO:0050679	positive regulation of epithelial cell proliferation	0.001681	5
GO:0050673	epithelial cell proliferation	0.002676	6
GO:0050730	regulation of peptidyl-tyrosine phosphorylation	0.002676	5
GO:0050769	positive regulation of neurogenesis	0.002676	6
GO:0042063	gliogenesis	0.002676	5
GO:0002833	positive regulation of response to biotic stimulus	0.002676	3
GO:0046850	regulation of bone remodeling	0.002676	3
GO:0001818	negative regulation of cytokine production	0.002676	5
GO:0030665	clathrin-coated vesicle membrane	0.01508	3
GO:0043235	receptor complex	0.01508	4
GO:0030662	coated vesicle membrane	0.0269	3
GO:0030136	clathrin-coated vesicle	0.0269	3
GO:0016323	basolateral plasma membrane	0.028616	3
GO:0008083	growth factor activity	7.32 × 10^−5^	5
GO:0048018	receptor ligand activity	0.000263	6
GO:0030545	receptor regulator activity	0.000263	6
GO:0004713	protein tyrosine kinase activity	0.010353	3
GO:0005154	epidermal growth factor receptor binding	0.012091	2
GO:0004715	non-membrane spanning protein tyrosine kinase activity	0.016435	2

Note: Green bar, biological process. Blue bar, cellular components. Pink bar, molecular function.

**Table 3 ijms-22-02442-t003:** The IRGs involved in the predictive model.

Gene	Coef ^1^	HR ^2^	HR.95L	HR.95H	*p*-Value
APOD	−0.06584	0.936277	0.847164	1.034765	0.196953
TFRC	0.004018	1.004026	0.999878	1.008191	0.057172
GRN	0.00648	1.006501	1.003542	1.009469	1.61E−05
CSK	−0.04999	0.951235	0.91998	0.983551	0.003357
HDAC1	−0.01997	0.980231	0.963732	0.997013	0.021143
NFATC4	0.129489	1.138247	1.011662	1.280671	0.03134
BMP6	0.055054	1.056598	0.992466	1.124874	0.084843
IL17RD	0.124096	1.132124	0.971935	1.318714	0.110877
IL3RA	−0.22745	0.79656	0.688675	0.921347	0.00219
LEPR	0.520483	1.68284	1.112013	2.546688	0.013808

^1^ Coef: coefficient; ^2^ HR: hazard ratio.

**Table 4 ijms-22-02442-t004:** Univariate and multivariate cox regression analyses.

Clinicopathological	Univariate Analysis			Multivariate Analysis		
Characteristics	HR	95% CI	*p*-Value	HR	95% CI	*p*-Value
Age > 45 years	1.048	0.608–1.807	0.865	–	–	–
Grade 3–4	1.043	0.597–1.822	0.883	–	–	–
FIGO stage			**0.001 ***			**0.016 ***
I	–	–	–	–	–	–
II–III	0.884	0.473–1.654	0.701	0.742	0.395–1.394	0.353
IVA	5.121	1.956–13.408	**0.001 ***	2.778	1.019–7.575	**0.046 ***
IVB	3.139	1.095–9.000	**0.033 ***	2.891	1.006–8.306	**0.049 ***
Histological type (squamous carcinoma vs. adenocarcinoma)	1.513	0.545–4.204	0.427	–	–	–
High risk	3.369	1.846–6.147	**0.001 ***	3.170	1.701–5.910	**0.001 ***

******p* < 0.05.

**Table 5 ijms-22-02442-t005:** The clinical data of three para-tumor samples.

Serial Number	Age	pathological Pattern	Survival Time	Survival State	Grade	TNM	FIGO
TCGA-HM-A3JJ	45	Squamous cancer	659 days	dead	G3	T1b1N1M0	IB1
TCGA-FU-A3EO	55	Adenocarcinoma	490 days	alive	G2	T2b1N0M0	IIB
TCGA-MY-A5BF	68	Squamous cancer	634 days	alive	-	T2a2N0M0	IIA2

**Table 6 ijms-22-02442-t006:** The characteristic of 212 clinical samples.

Characteristics	Number of Cases (%)
Histological type	
Adenocarcinoma	22 (10.4)
Squamous cancer	190 (89.6)
Age (year)	
≤45	100 (47.2)
>45	112 (52.8)
Grade	
1–2	122 (57.5)
3–4	90 (42.5)
T stage	
I	116 (54.7)
II-III	82 (38.7)
IVa	5(2.4)
IVb	9(4.2)
Survival status	
Alive	159 (75.0%)
dead	53 (25.0%)
Duration of disease (year)	
≤5	176 (83.0)
>5	36 (17.0)

## Data Availability

The authors confirm that the data supporting the findings of this study are available within the article and its Appendix A.

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
