# Peer review of "Immunogenomic Identification for Predicting the Prognosis of Cervical Cancer Patients"

_ijms, 2021, doi:10.3390/ijms22052442_

Round 1
Reviewer 1 Report
Thank you for doing appropriate changes. I have only a few minor comments:
Materials and Methods: Are all these histopathological data available for all patients? If not, please comment in the text. You have not reported on the percentage of stromal versus squamous or glandular cells within the three normal samples. If this is not available, this should be stated. If the cells are mainly stromal, tissue specific instead of cancer specific differences could be what you see.
Author Response
Reviewer 1:
Comments: Materials and Methods: Are all these histopathological data available for all patients? If not, please comment in the text. You have not reported on the percentage of stromal versus squamous or glandular cells within the three normal samples. If this is not available, this should be stated. If the cells are mainly stromal, tissue specific instead of cancer specific differences could be what you see.
Response: Thank you for the questions. The histopathological data of 212 cervical cancer cases with complete clinical data are included in all these histopathological data. We only used all the histopathological data to screen out differently expressed genes, differently expressed immune-related genes, and differently expressed transcription factors. As to prognosis-related analysis, all analysis are performed with 212 cervical cancer cases with complete clinical data. To clarify the problem, we have emphasized in the part of material and method:
4.2. Analysis of DEGs
To identify the DEGs between three para-tumor and 289 cervical tumor tissue specimens, an R language script was made, and a limma package (http://www.bioconductor.org/packages/release/bioc/html/limma.html) was downloaded and performed by the R software.
4.3. Screening of differentially expressed SIRGs
IRGs significantly associated with survival of 212 tumor cases were identified from all the differentially expressed IRGs through univariate cox analysis (p<0.05). These IRGs were defined as SIRGs. Forest map script was performed by R software.
As to the percentage of stroma cells of the three normal samples, only could we find that of TCGA-FU-A3EO. The percentage of stroma cells of TCGA-FU-A3EO is 0. However, normal cervical cells are 100%. Therefore, the cells are mainly cervical cells. We have restated the problem as following:
4.1. Gene expression data and clinical data collection
As to para-tumor slides, the percentage of tumor cells, lymphocytes, necrosis, infiltrated monocytes, infiltrated neutrophil was 0, and the percentage of normal cells was 100%. The percentage of stroma cells of TCGA-FU-A3EO was 0, while that of the other two para-tumor tissue specimens was missing.

Reviewer 2 Report
This study identified immunogenomic for predicting prognosis and developed prognostic model based on ten immune-related genes.
Major comments
1. The authors compared expression of immune-related genes between cervical tumor and para-tumor tissues. However, they used 289 samples of cervical tumor and 3 of para-tumor tissues. So, there was a very great difference sample size between two groups to conclude the results. So, it may be better to identify the prognostic biomarker of immune-related genes based on oncological outcomes (recurrence or death).
Author Response
Reviewer 2:
Comments: 1. The authors compared expression of immune-related genes between cervical tumor and para-tumor tissues. However, they used 289 samples of cervical tumor and 3 of para-tumor tissues. So, there was a very great difference sample size between two groups to conclude the results. So, it may be better to identify the prognostic biomarker of immune-related genes based on oncological outcomes (recurrence or death).
Response: Thank you for the question. Yes, you are right. Indeed, we only used all the histopathological data to screen out differently expressed genes. As to prognosis-related analysis, including the prognostic biomarker, all analysis are performed with 212 cervical cancer cases with complete overall survival data whose oncological outcome is death.
As to the language changes, we tried our best to improve the manuscript and made some changes in the manuscript. These changes will not influence the content and framework of the paper. And here we did not list the changes but marked in red in revised paper. We appreciate for Editors/reviewers’ warm work earnestly, and hope that the correction will meet with approval. Once again, thank you very much for your comments and suggestions.

Round 2
Reviewer 2 Report
There is no comment.
This manuscript is a resubmission of an earlier submission. The following is a list of the peer review reports and author responses from that submission.
Round 1
Reviewer 1 Report
In this study, the authors have applied the TCGA dataset to identify 204 differentially expressed immunogens between a cohort of 286 cervical carcinomas and three para-tumor samples. Further, they performed COX analysis, which identified 20 genes as significantly associated to survival. From these 20 genes, 10 genes were selected for inclusion in the prediction model. The authors conclude that this prediction model has moderate accuracy and stability for predicting prognosis in early stage squamous cell carcinoma.
General comments:
This paper needs to be restructured. The results are not presented in a way that makes it easy to follow. The results are not properly discussed and the discussion should include a paragraph stating study limitations. The methods do not contain enough information on which analyses and cut offs that have been used. The figures are poorly presented. The language is poor and needs an extensive editing. Please see specific comments below.
Abstract:
Line 18: Please explain how the tumor micro environment functions in cervical cancer.
What do you mean with treated more positively? Please rephrase!
Introduction:
Please decide of cervical cancer is mainly (as you state in the introduction) or solely (as you state in the abstract) caused by HPV.
Line 40: Please rephrase the second sentence of the introduction. It is not grammatically correct.
Line 44: Please rephrase ‘as much as’
Line 56: The use of ‘until’ seems to be wrong. Perhaps ‘With’ is what you mean?
Line 72: Please correct the typo.
Results:
Please explain more thoroughly which samples you are comparing in each analysis. Please describe the para-tumor cohort. From where are the para-tumors collected? Have the tumor content in these samples been verified? Does the tumor samples from these three para-tumor samples exist? If so, it would be interesting to see an (e.g.) t-SNE plot displaying how closely the tumor samples are to their paired para-tumor sample. This is very statistically relevant when using a cohort of only three samples in comparison.
Line 100-101: Please explain better how the ROC curve explains that the JAK-STAT signaling pathway was the most frequently identified KEGG pathway.
The paragraph 2.2 needs to be restructured to better explain what was done. As an example, you should explain that among the 22 (or was it 20?) DEGs that you picked out as significantly associated with survival, TYK2, CSK, PTPN6 and IL3RA were hub genes (if this is correct).
Line 116: Please explain how you screened out the 12 genes. Also, in Table 1, 10 genes are displayed, but in the text 12 genes are mentioned. Please explain this.
Paragraph 2.5.: Please state if any other correlation to clinicopathological features was found for any of the other IIRGs. You should account for multiple testing here.
Line 162: Please explain the aberration SIRG the first time it is mentioned in the text (not just in figure legends).
Line 163: How did you select the 74 TFs?
Discussion:
The discussion should include a critical comment on the para-tumor collection, which only includes three samples. This is a very small sample size for doing differential gene expression analyses and makes the results prone to coincidences. Further, the authors should comment on whether they performed multiple testing when identifying genes with significant relations to survival and other clinicopatholgical features (FIGO stage, grade, age etc.). I see no FDR values for these analyses. Also, the authors should comment on whether there is a potential clinical application of these findings.
Line 176: Please use another word than developed. I would suggest advanced stage, late-stage or metastatic/beyond the cervix. Developed could mean once the lesions develops into cancer (early stage).
Line 197: Please explain why a predictive model is assumed to be better or refer to a study that shows or reviews this.
Line 204-265: This paragraph needs to be significantly decreased. Please focus on main findings, and highlight 2 to 5 of the SIRG genes. Instead of including all these other studies, you should focus on how your findings contribute to the characterization of high risk versus the low risk tumors.
Methods:
Please explain the criteria used for selecting the 20 IRG genes associated to survival and the 10 prediction model genes. Did you account for multiple testing? How did you perform COX analysis on 204 genes? Please go more into detail on this. Please explain which analyses were performed on which samples. When were the adenocarcinomas and the stage>I squamous cell carcinomas applied? Please explain what you mean by disease type. If you mean histology, please rephrase to ‘Histological type’.
Figures:
The layout of the figures needs to be significantly improved. Please put the different plots that you display in meaningful context. What are you trying to tell by each individual plot, are some of them redundant? Make sure that when you refer to a figure (e.g. Fig 2c), you describe what is actually shown in the figure (in the case of figure 2c, a survival plot, not a forest plot).
Figure 1: The heat maps in figure 1a and 1B are too small to give any relevant info and should be considered as supplementary material showing the whole heat map as in one page. Figure 1e is a table and should not be pasted in as part of a figure.
Figure 2: Please do not show so many Kaplan Meier plots of different subgroups. This should be put in supplementary. Figure 2a is not a Kaplan Meier plot, but a table. Figure 2b does not contain a ROC curve as is stated both in text and figure legends. Figure 2h should be enlarged for readability.
Tables:
Table 2: Please explain (for instance in footnotes) what you mean by ‘Disease type’.
Author Response
Reviewer 1;
General comment: This paper needs to be restructured. The results are not presented in a way that makes it easy to follow. The results are not properly discussed and the discussion should include a paragraph stating study limitations. The methods do not contain enough information on which analyses and cut offs that have been used. The figures are poorly presented. The language is poor and needs an extensive editing.
Response: Thank you for your professional comment on our article, which help us make progress in paper writing. According to your comments, we have made extensive modifications to our manuscript. We have reordered the figures, make up the inefficient description to the methods and results. We rewrote the discussion to make it focus on our topic. We are so sorry for the poor language. A native speaker has helped us to revise the language.
- comment: Abstract: Line 18: Please explain how the tumor micro environment functions in cervical cancer. What do you mean with treated more positively? Please rephrase!
Response: Thank you for your question. Tumor micro environment (TME) is an environment around a tumor. TME is consisted of primary tumor-derived components, tumor-mobilized bone-marrow-derived cells (BMDCs), and the local stromal microenvironment of the host, which is necessary for tumor metastasis. Before a circulating tumor cell seed to a new site, some molecular from primary tumor cells educate the new site first by paracrine or secreting vesicles. Moreover, the molecular can also recruit and polarize the BMDCs to pro-oncogenesis cells with immunosuppression and educate the local stromal microenvironment to angiogenesis, tissue remodel. Finally, the new site is equipped with a proper condition for the circulated tumor cell to seed and proliferate[1]. Therefore, tumor micro environment is necessary for tumor metastasis, and cervical caner cell is no exception. For example, cervical cancer derived molecular CCL2 could recruit monocyte[2-3] and IL-10 could polarize monocytes to M2 macrophages which is immunosuppressive[4].
Thank you for pointing out the vague word. I would like to change it to be “strictly”. I mean, NCCN cervical cancer clinical practice guideline 2019 shows that In the early stage of cervical cancer treatment, surgery was the main treatment, in the middle and late stage, radiotherapy was the main treatment, supplemented by chemotherapy. Meanwhile, doctors should also pay attention to individualized treatment, such as taking age, specific immune targets and general condition into consideration. Since the early-stage cervical squamous patients with high risk score represents worse prognosis according to the prognostic model, and prognosis is closely associated with clinical stage, pathological pattern and therapies, we should adopt more strict therapies to the patients with high risk score. For example, for patients with FIGO IA2-IIA, we usually adopt extensive hysterectomy and pelvic lymph node dissection. Para-aortic lymph node dissection or sampling is performed when necessary. While for patients with high risk score, maybe we should perform para-aortic lymph node dissection or sampling, which needs more clinical data to verify. For patients with FIGO I A1-IB1 whose tumor diameter less than 2cm and who hope to be pregnant, we usually adopt extensive cervectomy and pelvic lymph node dissection. While for patients with high risk, maybe we should adopt extensive hysterectomy and pelvic lymph node dissection to decrease the recurrence probability. What is more, for patients with high risk score, we should increase the frequency of follow-up.
Reference
[1] Liu Y, et al., Cancer Cell. 2016. Nov 14;30(5):668-681.
[2] J. C. Pahler et al., Neoplasia. Plasticity in tumor-promoting inflammation: impairment of macrophage recruitment evokes a compensatory neutrophil response. 2008. 10, 329-340
[3] E. Sierra-Filardi et al., J Immunol. 2014. CCL2 shapes macrophage polarization by GM-CSF and M-CSF: identification of CCL2/CCR2-dependent gene expression profile. 192, 3858-3867 [4] J. Kim, J. S. Bae, Mediators Inflamm. 2016. Tumor-Associated Macrophages and Neutrophils in Tumor Microenvironment. 2016, 6058147
- comment: Introduction: Please decide of cervical cancer is mainly (as you state in the introduction) or solely (as you state in the abstract) caused by HPV. Line 40: Please rephrase the second sentence of the introduction. It is not grammatically correct. Line 44: Please rephrase ‘as much as’. Line 56: The use of ‘until’ seems to be wrong. Perhaps ‘With’ is what you mean? Line 72: Please correct the typo.
Response: Thanks for your comments. We feel really sorry for our carelessness. I would use mainly because a fraction of cervical cancer is attributable to HIV, hypo immunity, smoke. Line 40: rephrased sentence: There are approximately 527,000 new cases of cervical cancer appear and 265,000 related deaths annually. Line 44: rephrased words: up to. Line 56: rephrased word: with. Line 72. We have changed the format to full-justified.
- comment: Results: Please explain more thoroughly which samples you are comparing in each analysis. Please describe the para-tumor cohort. From where are the para-tumors collected? Have the tumor content in these samples been verified? Does the tumor samples from these three para-tumor samples exist? If so, it would be interesting to see an (e.g.) t-SNE plot displaying how closely the tumor samples are to their paired para-tumor sample. This is very statistically relevant when using a cohort of only three samples in comparison.
Response: Thank you for your question. In the part of result 2.1 that Identification of differentially expressed genes, the changes are as following: A total of 2,240 DEGs were identified by comparing the gene expression data between three para-tumor tissue specimens and 286 primary cervical tumor tissue specimens, including 1412 down-regulated and 1,928 up-regulated ones (Fig. S1a and 1a). By comparing the IRGs obtained from the Immunology Database and Analysis Portal (ImmPort) database[12] with the DEGs, 204 IRGs overlapped with DEGs were selected. In the part of result 2.2 that Identification of differentially expressed IRGs associated with survival, the change is as following: First, we identified 20 differentially expressed IRGs associated with survival. In the part of result 2.3. A total of 10 IRGs were screened out using the multivariate COX analysis, which were independent factors for the survival state and time of 212 cases with clinical data. 2.4. Verifying the efficacy of the prognostic model, the change is as following: To verify the efficacy of the prognostic model, we performed a Kaplan-Meier survival analysis for the 212 cases. In the part of result 2.5 The clinical significance of IIRGs, the change is as following: The differences of IRGs in clinicopathological characteristics including age, grade, FIGO stages, of the 212 cases were analyzed to determine the relationship between the risk score and clinical parameters.
As to the para-tumor cohort, there are three para-tumor cases. One is squamous cervical cancer coded TCGA-HM-A3JJ. The survival time is 659 days. Survival state is dead. Age is 40 years old. Grade is G3. TNM staging is T1b1N1M0. FIGO stage is IB1. The second is adenocarcinoma coded TCGA-FU-A3EO. The survival time is 490 days. Survival state is alive. Age is 55 years old. Grade is G2. TNM staging is T2b1N0M0. FIGO stage is IIB. The third one is squamous cervical cancer coded TCGA-MY-A5BF. The survival time is 634 days. Survival state is alive. Age is 68 years old. TNM staging is T2a2N0M0. FIGO stage is IIA2. We have made a table for describing the information.
Among the 212 samples with clinical data, 150 cases, 14 cases, 26 cases and 6 cases are from the white race, the Asian, the black or African American, American Indian or a laska native respectively. And 16 cases’ records about race are lost.
Yes, the percentage of tumor cells are verified. The range is from 60% to 95%.
Indeed, as to the study about the role of immune in cancer, it is important to know the relative location between immunocytes and cancer cell. Therefore, your idea is deserved to be performed in the future studies by Spatial transcriptomics. However, there is no overlap between the tumor samples and para-tumor samples. Moreover, this data obtained from TCGA has no spatial information.
Comment: Line 100-101: Please explain better how the ROC curve explains that the JAK-STAT signaling pathway was the most frequently identified KEGG pathway.
Response: Thank you for your reminding. We feel really sorry for the carelessness. We have changed the image of ROC curve to the image of KEGG pathway.
Comment: The paragraph 2.2 needs to be restructured to better explain what was done. As an example, you should explain that among the 22 (or was it 20?) DEGs that you picked out as significantly associated with survival, TYK2, CSK, PTPN6 and IL3RA were hub genes (if this is correct).
Response: Thank you for your nice comments on our article. According to your suggestions, what we have corrected are shown below: Since survival time and status are important to the prognostic evaluation, it seems to be feasible to evaluate the prognosis of patients based on the expressions of genes associated with survival. First, we identified 20 differentially expressed IRGs associated with survival. GO analysis revealed that tumor-related biological process was the most frequently implicated term (Table 2). JAK-STAT signaling pathway was the most frequently identified KEGG pathway (Fig. 2a). A forest plot of hazard ratios indicated that 10 genes were significant protective factors, and 12 genes were significant adverse factors (Fig. 2b). Protein-protein interaction (PPI) network analysis demonstrated that TYK2, CSK, PTPN6, and IL3RA were the hub genes, which were screened out based on the criteria of correlation coefficient 0.3 and the number of interactive genes no less than 3 (Fig. 2c). These hub genes were actively involved in the JAK-STAT signaling pathway (Fig 2d).
Comment: Line 116: Please explain how you screened out the 12 genes. Also, in Table 1, 10 genes are displayed, but in the text 12 genes are mentioned. Please explain this.
Response: Thank you for your question. We adopted Surv () function to do multivariate cox regression analysis performed by R. The R package of Surv (time, event) function was downloaded.
We are sorry for making you confused because of our carelessness. It should be 10 genes in the text in accordance with the data in the table.
Comment: Paragraph 2.5.: Please state if any other correlation to clinicopathological features was found for any of the other IIRGs. You should account for multiple testing here.
Response: Thank you for your question. Apart from those three IIRGs, there are no other IIRGs are correlated significantly with the clinicopathological features.
We get SIRGs by univariate cox regression. Each SIRG is the independent factor to survival. We perform multivariate cox regression to get IIRGs, which means we get a group of IIRGs and they will act as an independent factor to survival synergistically after age, histological type, grade and FIGO stage have been adjusted.
Comment: Line 162: Please explain the aberration SIRG the first time it is mentioned in the text (not just in figure legends).
Response: We are so sorry for making such careless fault. When we replicated the figures from original text to the template, we replicated the wrong figures that should have been in Figure 2 but the legend was replicated correctly. That is the reason that the aberration happens.
Comment: Line 163: How did you select the 74 TFs?
Response: Thank you for your question. The list of transcription factors (TFs) related to cancer was downloaded from the Cistrome database (http://cistrome.org/). And then we intersected the list of TFs and the DEGs and obtained the differentially expressed 74 TFs (DETFs).
4.Comment: Discussion: The discussion should include a critical comment on the para-tumor collection, which only includes three samples. This is a very small sample size for doing differential gene expression analyses and makes the results prone to coincidences. Further, the authors should comment on whether they performed multiple testing when identifying genes with significant relations to survival and other clinicopathological features (FIGO stage, grade, age etc.). I see no FDR values for these analyses. Also, the authors should comment on whether there is a potential clinical application of these findings.
Response: We feel great thanks for your professional review work on our article. As you are concerned, there are several problems that need to be addressed. According to your nice suggestions, we have pointed out the limitation in the last paragraph of discussion: We must point out that only three control specimens were acquired in the present study. Although it met the minimum requirements for biological repeat, insufficient control samples tended to cause larger errors. Therefore, more other experiments are still necessary to validate the transcriptome results.
We did not perform multiple testing when identifying genes with significant relations to survival. We adopted multivariate cox regression. And as to identifying genes with significant relations to clinicopathological features, we analyzed the gene expression of single gene in two independent groups by independent t test.
Thank you for your suggestion. First, from the Fig. 3a, we could tell that in terms of high-risk group, the probability of 5 year survival is about 55%, while that of low-risk group is about 80%. Therefore, we could tell the probability of survival by our model. Second, we supplemented a result 2.6. The correlation between IIRGs and immunocytes infiltration: Immune infiltration plays an important role in tumor progression and prognosis. To deeply explore the IIRGs, we analyzed the correlation between 10 IIRGs and immunocyte infiltration in cervical cancer using the Timer database. The criteria of |coefficient| >0.3 and p<0.05 were used as correlatives. Figure S3 shows that GRN was significantly positively correlated with the infiltration of neutrophils and dendritic cells. CSK wa significantly positively correlated with the infiltration of CD4+ T cells. NFATC4 was significantly positively correlated with the infiltration of macrophages. BMP6 was significantly positively correlated with the infiltration of macrophages. IL3RA was significantly positively correlated with the infiltration of B cells, CD4+T cells, macrophages and dendritic cells. Interestingly, all the significantly correlated IIRGs were positively correlated. Therefore, high-risk scores may suggested more immunocyte infiltration.
Comment: Line 176: Please use another word than developed. I would suggest advanced stage, late-stage or metastatic/beyond the cervix. Developed could mean once the lesions develops into cancer (early stage).
Response: Thank you for your advice. We change the sentence to “Presently, although the development of surgery, radiation therapy, and chemotherapy, the rates of recurrence and metastasis in patients with late-stage cervical cancer are still up to 40.3% and 31%, respectively”.
Comment: Line 197: Please explain why a predictive model is assumed to be better or refer to a study that shows or reviews this.
Response: Thank you for your question. I would like to refer a similar study named A prognostic signature of five pseudogenes for predicting lower-grade gliomas (Biomed Pharmacother. 2019 Sep;117:109116. PMID: 31247469). The molecular consisting the predictive model are associated with prognosis of cervical cancer significantly. Therefore, they are potential clinical prognostic biomarkers or therapeutic targets that needs further experiments to verify. Moreover, the prognostic model performed moderately and steady in cervical squamous patients with FIGO stage I regardless of the age and grade. we could tell the probability of survival by our model. Finally, it is convenient to obtain the samples along with surgery, and the second generation of RNA-seq is common to be used in hospitals.
Comment: Line 204-265: This paragraph needs to be significantly decreased. Please focus on main findings, and highlight 2 to 5 of the SIRG genes. Instead of including all these other studies, you should focus on how your findings contribute to the characterization of high risk versus the low risk tumors.
Response: We feel great thanks for your professional review work on our article. As you are concerned, there are several problems that need to be addressed. According to your nice suggestions, we have made extensive corrections to our previous draft. We streamed the information about SIRGs. We supplemented the discussion about prognostic model and immunocytes infiltration. The detailed corrections are in yellow below.
Cervical cancer is caused by the persistent infection of hrHPV [15]. Although surgery, chemoradiotherapy, anti-angiogenic medicine and even the new immunotherapy have been applied for cervical cancer treatment, the prognosis remains poor at the late stage [4, 9, 16]. Therefore, research on effective prognostic biomarkers and new molecular mechanisms has drawn increasing attention. Kidd EA et al. have found that the standardized uptake value for F-18 fluorodeoxyglucose is a sensitive predictive biomarker for the survival of cervical cancer patients [17]. Luo W et al. have identified a 6 lncRNAs signature, which can be regarded as novel diagnostic biomarkers for cervical cancer [18]. Li X et al. have identified a histone family gene signature for predicting the prognosis of cervical cancer patients [19]. These studies provide an elemental knowledge of the pathogenesis of cervical cancer at the genetic level. However, the prognostic role of immunogenomics in cervical cancer remains largely undetermined. In the present study, we performed a comprehensive analysis of IRGs in cervical cancer, which might enhance our knowledge of their clinical value and help us understand potential molecular mechanisms. Moreover, these IRGs might act as valuable clinical biomarkers or therapeutic targets. Besides, we constructed a prognostic model that could help assess immune cell infiltration and potential clinical outcomes of cervical cancer patients.
Tumor immune microenvironment can promote the progression of cervical cancer, including cancer cell proliferation, invasion, metastasis, immunosuppression and tissue remodeling, fibrosis, and angiogenesis. For example, CXCL12 induces mononuclear phagocytes to release HB-EGF, triggering anti-apoptotic and proliferative signals in Hela cells [20]. D-dopachrome tautomerase (D-DT), a homolog of macrophage migration inhibitory factor (MIF), can promote the invasion of cervical cancer cells when it is overexpressed [21]. An altered balance in IL-12p70 and IL-10 production can weaken T cell proliferation in cervical cancer [22]. These studies suggest the importance of immunity in cervical cancer progression. Therefore, it is necessary to identify differentially expressed IRGs. Genome profile alterations cause tumorigenesis. We identified alternations in immunogenomic profiles to study the effect of alternations on the immune microenvironment and clinical prognosis. Gene functional enrichment analysis suggested that these genes were mainly involved in growth factor (GF) activity. GFs actively act in the pathogenesis of cervical cancer. Notably, these GFs are correlated to proliferation, aggression and migration [23,24,25]. Therefore, these GFs could also be used to monitor metastasis, assessing survival, and identify potential drug targets as clinical biomarkers.
Among the 20 SIRGs, no related reports have explored the roles of DUOX1, GRN, CSK, CD79, NFATC4, EPGN, TGFA, IL17RD, LEPR, N2RF1 and TRAV26-1 in cervical cancer. APOD is down-regulated in cervical cancer compared with normal cervix[26]. TFRC expression is up-regulated in cervical cancer compared with normal cervix[27]. A significant correlation between cervical cancer and the polymorphism of rs1041981 in the LTA gene has been observed[28]. The rare allele (A) of SNP rs2239704 in the 5′ UTR of the LTA gene is significantly associated with increased risks of cervical cancer[29]. F2RL1 is overexpressed in cervical cancer cell lines and significantly correlated with poor OS[30,31]. HGF overexpression in lesions of cervical cancer has been reported to be related to a poorer prognosis[32]. HDAC1/DNMT3A-containing complex is associated with the suppression of cancer stem cells in cervical cancer [33]. HGF can induce migration and invasion of cervical cancer cells [34].] BMP6 may participate in invasion and metastasis in cervical cancer [35]. The proportion of CD123(+) dendritic cells is significantly lower in the peripheral blood of cervical cancer patients compared with the controls [36]. A higher frequency of Nrp1(+) T-regs frequency suppress the immune response against distant cervical cancer cells [37]. The above-mentioned results are consistent with our current findings. However, Tyk2 is confirmed to be overexpressed in SCC[38], which was different from our study. Low-throughput experiments, such as Western blotting analysis, is required to verify the factual expression. PTPN6 is positively correlated with HPV infection in cervical cancer with the explanation of cell defense reaction [39].
To explore molecular mechanisms underlying the potential clinical importance, we constructed a TF-mediated network that could regulate hub IRGs. Among the SDETFs, Foxp3 significantly associated with FIGO stage and tumor size [40]. Foxp3 is associated with lymphangiogenesis of cervical cancer[41]. FoxP3 has been confirmed to be highly expressed in cervical cancer, and it facilitates the proliferation and invasiveness and inhibits the apoptosis of cervical cancer cells [42]. In conclusion, Foxp3 is a risk factor for the survival of cervical cancer, which is consistent with our current finding. CBX7 inhibits the proliferation of cervical cancer cells [43]. LTA inhibits the proliferation of CD4(+) T-cells in a FoxP3(+) Treg-dependent manner in patients with chronic hepatitis C, suggesting that LTA acts on FoxP3[44]. Therefore, previous studies provide limited information about the mechanisms of 10 IRGs in the survival of cervical cancer.
The effects of the JAK/STAT pathway and the persistent activation of STAT3 and STAT5 during the process of tumor cell proliferation, cycling and invasion have made it a favorite treatment target. In cervical cancer, the activated JAK/STAT signaling pathway by Bcl-2 promots cell viability, migration, and invasion [45]. There is a strong association between HPV infection and STAT-3 overexpression in cervical cancer [46]. The expression of STAT3 has been proposed as a poor prognostic factor in cervical cancer [47]. STAT5 protein is up-regulated and associated with the severity of cervical cancer [48]. Moreover, overexpression of STAT-5 elevates the STAT-3 expression compared with the normal controls [46]. Therefore, JAK/STAT signaling may play an important role in cervical carcinogenesis. Moreover, up-regulation of PAR2 (F2RL1) induces the proliferation of cervical cancer cells by activating STAT3 [49], which is consistent with our current results that overexpression of F2RL1 was involved in the JAK/STAT signaling pathway.
In the present study, we created an immune-based prognostic signature to monitor the immune status and access the prognosis for cervical cancer patients. Previously, Wu HY et al. (2020) have constructed a prognostic index based on percent-splice-in values in SCC [50]. Eun Jung Kwon et al. (2020) have explored genomic alterations and developed a risk index model can monitor HPV- related bladder cancer [51]. Cai LY has created a risk score model based on differentially expressed glycolysis-related genes, and the model can predict the prognosis of cervical cancer patients [52]. Recently, Zhao S et al. (2020) have constructed a 4-gene prognostic risk score model in CESC by identifying DEGs [53]. Beyond the above-mentioned studies, there are also many studies about prognostic model [54,55]. Compared with the previous studies, our prognostic model could also assess the immunocyte infiltration based on not only genomic alterations but also immune-genomic profiles. Moreover, we constructed a Tf-mediated regulatory network, which provided a more detailed mechanism of IIRGs. Our prognostic index, based on 10 differentially expressed IIRGs in cervical cancer, demonstrated favorable clinical viability. Our data showed that the risk score model performed moderately and steadily in prognostic predictions in patients with early-staged cervical cancer patients.
Among these 10 IIRGs, five IIRGs were significantly correlated with immunocyte infiltration, especially neutrophils, dendritic cells, CD4+ T cells, macrophages, and B cells, indicating that a greater immunocyte infiltration might be observed in low-risk patients. Our results confirmed immunocyte infiltration promoted tumor progression. Many studies have investigated the role of immunocytes in cervical cancer. Zhou PJ et al. (2020) have pointed out a high neutrophil to- lymphocyte ratio is independently associated with decreased OS and PFS in patients with cervical cancer by meta-analysis [56]. Ma Y et al. (2013) have shown that dendritic cells exhibit tolerogenic or immunosuppressive states that favor malignant progression[57]. K Manjgaladze et al. (2019) have proved that the infiltration of both CD4 T helper cells and CD8 cytotoxic T cells is significantly increased during the progression of cervical intraepithelial neoplasia [58]. M2-like macrophages are increasingly infiltrated in cervical cancer [59]. O'Brien PM et al. (2001) have shown that B-lymphocytes are the predominant lymphocyte infiltrate in pre-malignant cervical lesions to compromise the host immune response [60]. These results are consistent with our conclusions. However, the role of immune cells in cervical cancer still remains largely unclear. Further research is needed.
We must point out that only three control specimens were acquired in the present study. Although it met the minimum requirements for biological repeat, insufficient control samples tended to cause larger errors. Therefore, more other experiments are still necessary to validate the transcriptome results.
- Comment: Methods: Please explain the criteria used for selecting the 20 IRG genes associated to survival and the 10 prediction model genes. Did you account for multiple testing? How did you perform COX analysis on 204 genes? Please go more into detail on this. Please explain which analyses were performed on which samples. When were the adenocarcinomas and the stage>I squamous cell carcinomas applied? Please explain what you mean by disease type. If you mean histology, please rephrase to ‘Histological type’.
Response: Thank you for your question. We adopted univariate cox regression to screen the IRGs associated to survival (SIRGs) out. P<0.05 is the criteria.
As to the 10 prediction model genes, we adopted multivariate cox regression to analyze the 20 SIRGs. P<0.05 is the criteria. Because we did not adopt t-test to analyze data among multiple groups, adjusted p value is not necessary.
Yes, I have accounted for multiple test for the 4th comment: We did not perform multiple testing when identifying genes with significant relations to survival. We adopted multivariate cox regression. And as to identifying genes with significant relations to clinicopathological features, we analyzed the gene expression of single gene in two independent groups by independent t test.
As to the analysis on 204 genes, firstly we downloaded the R language package of survival and then we edit script by R language to run 204 recycles to perform the univariate COX regression for each IRG.
Thank you for your suggestion. We have supplemented the concrete information for analyzing data in the part of method.
We feel sorry for the lack of description about this part. The adenocarcinomas and the stage I squamous cell carcinomas were applied when we verify the efficacy of the prognostic model. We analyzed the difference of survival between high risk group and low risk group in adenocarcinoma cases and squamous carcinoma cases respectively. We have made up this part in method: Survival analysis was performed between the high-risk group and low-risk group in terms of age (age≤45,age>45), grade (1-2, grade3-4), cancer type (cervical adenocarcinoma and SCC), and FIGO stage (I, II-III, IVa and Ivb) by SPSS. A p value of less than 0.05 was considered statistically significant We are sorry for the inaccurate word. We have changed the word to histological type as you suggested.
- Comment: Figures: The layout of the figures needs to be significantly improved. Please put the different plots that you display in meaningful context. What are you trying to tell by each individual plot, are some of them redundant? Make sure that when you refer to a figure (e.g. Fig 2c), you describe what is actually shown in the figure (in the case of figure 2c, a survival plot, not a forest plot).
Response: We are so sorry for our careless mistakes. Thank you for your reminding. We have put the figure to right place. And we supplemented the information in the legend of figure3 and figure 4.
Comment: Figure 1: The heat maps in figure 1a and 1B are too small to give any relevant info and should be considered as supplementary material showing the whole heat map as in one page. Figure 1e is a table and should not be pasted in as part of a figure.
Response: Thank you for your suggestion! We have make figure 1a and 1b supplementary materials.
Comment: Figure 2: Please do not show so many Kaplan Meier plots of different subgroups. This should be put in supplementary. Figure 2a is not a Kaplan Meier plot, but a table. Figure 2b does not contain a ROC curve as is stated both in text and figure legends. Figure 2h should be enlarged for readability.
Response: Thank you for your suggestion! We have made the Kaplan Meier plots of different subgroups in supplementary.
We are sorry for the careless mistake. We have put the right figure to right place.
We have enlarged the figure.
Comment: Tables: Table 2: Please explain (for instance in footnotes) what you mean by ‘Disease type’.
Response: We are sorry for the inaccurate description. We have changed the disease type to Histological type (squamous carcinoma vs. adenocarcinoma) in the table.

Reviewer 2 Report
This study identified immunogens for predicting prognosis and developed prognostic model based on ten immungenes.
Major comments
- In this study, TCGA public database was used to screen out the immunogens and transcription factors for predicting prognosis. Why did you analysis the differentially expressed genes between para-tumor and cervical tumor tissues for predicting prognosis. In general, the differentially expressed genes are compared between recurrent vs. non-recurrent groups or death or live groups for predicting prognosis.
- Could you provide clinical and tumor characteristics of TCGA public database?
- What was endpoint of prognostic model? recurrence or death?
- The authors developed prognostic model based on ten immungenes using TCGA public database. Moreover, prognostic model was validated using clinical data of 212 cervical cancer. Could you explain more detail methods of validation using clinical data? Did you analysis immunogens to divide low and high-risk groups?
Author Response
Reviewer 2:
- Comment: In this study, TCGA public database was used to screen out the immunogens and transcription factors for predicting prognosis. Why did you analysis the differentially expressed genes between para-tumor and cervical tumor tissues for predicting prognosis. In general, the differentially expressed genes are compared between recurrent vs. non-recurrent groups or death or live groups for predicting prognosis.
Response: Thank you for your question! We firstly identified the differentially expressed genes by comparing genes in cervical tumor with genes in para-tumor tissue. And then we selected out the prognostic genes by performing univariate cox regression between live group and dead group among tumor cases. Therefore, what you suggest is consistent with what we did.
- Comment: Could you provide clinical and tumour characteristics of TCGA public database?
Response: Thank you for your question. The clinical and tumour characteristics of TCGA public database is the following. Moreover, the original data of clinical and tumour will be attached in the email. We abandoned the data whose survival time less than 90 days during analysis.
Table 3. The Clinical Data of 3 Para-tumor Samples
|
Serial number |
Age (year) |
pathological pattern |
Survival time (day) |
Survival state |
Grade |
TNM |
FIGO |
|
TCGA-HM-A3JJ |
45 |
Squamous cancer |
659 |
dead |
G3 |
T1b1N1M0 |
IB1 |
|
TCGA-FU-A3EO |
55 |
Adenocarcinoma |
490 |
alive |
G2 |
T2b1N0M0 |
IIB |
|
TCGA-MY-A5BF |
68 |
Squamous cancer |
634 |
alive |
--- |
T2a2N0M0 |
IIA2 |
Table 4. The Characteristic of 212 Clinical Samples
|
Characteristics |
Number of cases (%) |
|
Histological type |
|
|
Adenocarcinoma |
22 (10.4) |
|
Squamous cancer |
190 (89.6) |
|
Age (year) |
|
|
<=45 |
100 (47.2) |
|
>45 |
112 (52.8) |
|
Grade |
|
|
1-2 |
122 (57.5) |
|
3-4 |
90 (42.5) |
|
T stage |
|
|
I |
116 (54.7) |
|
II-III |
82 (38.7) |
|
IVa |
5(2.4) |
|
IVb |
9(4.2) |
|
Survival status |
|
|
alive |
159 (75.0%) |
|
dead |
53 (25.0%) |
|
Duration of disease (year) |
|
|
<=5 |
176 (83.0) |
|
>5 |
36 (17.0) |
- Comment: What was endpoint of prognostic model? recurrence or death?
Response: Thank you for your question. The endpoint of prognostic model is death. We are so sorry that there is no recurrence time found in the original clinical data.
- Comment: The authors developed prognostic model based on ten immune genes using TCGA public database. Moreover, prognostic model was validated using clinical data of 212 cervical cancer. Could you explain more detail methods of validation using clinical data? Did you analysis immunogens to divide low and high-risk groups?
Response: Thank you for your question. We performed a multivariate cox regression to select out ten immune genes which are independent influencer to the survival to construct a prognostic model which is risk score=the plus of each gene expression multiply coefficient. The median of risk score of 212 cervical cancer is the cut off value of high risk group and low risk group. And then we validated the efficiency of the model. Firstly, we analysed the survival difference between high risk group and low risk group. The significant difference showed that the prognostic model could distinguish different clinical outcome from cervical cancer patients. In addition, we also performed ROC. The area under curve was 0.738 between 0.7 to 0.9, suggesting moderate accuracy for the prognosis in cervical cancer. Moreover, the model could distinguish the prognosis in subgroups of age and grade, while it can only work in squamous cervical cancer and FIGO I stage, which suggested that the model is steady in squamous cervical cancer patients with early stage regardless the age or grade. Risk curve showed that patients could be divided into high risk group and low risk group by the median of risk score, the survival time of high risk group was lower than that of low risk group. Moreover, the number of the death was more in high risk group than that in low risk group. The differentially expressed genes involved in the prognosis model was shown in the heat map which was consistent with the trend which the correlation coefficients showed. Univariate analysis and multivariate analysis indicated that the risk score could be an independent predictor (HR=3.170, 95%CI[ 1.701-5.910], p=0.001) after being adjusted by other clinicopathological characteristics such as age, grade, FIGO stage. Therefore, the predicting model could be a reliable and steady method to judge the clinical outcome for the squamous cervical cancer patients with FIGO I stage.

Reviewer 3 Report
The manuscript would strongly benefit from editing by a native speaker of English.
In the introduction the authors refer to data from a clinical phase II trial despite that Pembrolizumab has FDA approval for cervical cancer since 2018 and is widely used for treatment of qualifying cervical cancer patients (companion PD-L1 IHC 22C3 pharmDx test). Generally, explanation of checkpoint inhibitors as treatment options would be more appropriate than examples of mouse models.
In the study design, what are “Paratumor tissues”, how were they selected and defined? Is this cervical stroma, paracervical soft tissue, or something else?
The manuscript needs to better explain how and why the specific 204 IRGs were selected. This also applies to the 12 IRGs used in the prognostic model. What are the IRGs defining the low-risk and the high-risk group, and how were they chosen?
The discussion repeats many elements from the introduction and should focus more on the identified IRGs in the analysis and their significance for predicting prognosis.
Author Response
Reviewer 3:
- Comment: The manuscript would strongly benefit from editing by a native speaker of English.
Response: We are so sorry for the poor English writing. We have revised the language as you suggested. The changed places are marked in yellow.
- Comment: In the introduction the authors refer to data from a clinical phase II trial despite that Pembrolizumab has FDA approval for cervical cancer since 2018 and is widely used for treatment of qualifying cervical cancer patients (companion PD-L1 IHC 22C3 pharmDx test). Generally, explanation of checkpoint inhibitors as treatment options would be more appropriate than examples of mouse models.
Response: Thank you for your suggestion. We have changed the introduction as you suggested:
Currently, the treatment for immune checkpoint, such as cytotoxic T lymphocyte 4 (CTLA-4), programmed death protein 1 (PD-1), and its ligand (PD-L1), has shown initial success against cervical cancer [9].
- Comment: In the study design, what are “Para-tumor tissues”, how were they selected and defined? Is this cervical stroma, paracervical soft tissue, or something else?
Response: Thank you for your question. Para-tumor tissue refers to the tissue 2cm away from the lesion. They have checked the slides about percentage of tumor cells, infiltrated monocytes and normal cells. As to para-tumor slides, the percentage of tumor cell is 0, infiltrated monocytes are 0, and normal cells are 100%.
- Comment: The manuscript needs to better explain how and why the specific 204 IRGs were selected. This also applies to the 12 IRGs used in the prognostic model. What are the IRGs defining the low-risk and the high-risk group, and how were they chosen?
Response: Firstly, we screened out the differentially expressed genes by comparing the para-tumor and tumor tissue. Then we downloaded the list of IRGs from the Immunology Database and Analysis Portal (ImmPort) database. By intersecting the IRGs obtained in ImmPort database[12] with the DEGs, 204 IRGs overlapped with the DEGs were selected out.
The importance of immunology in cervical cancer has been established, molecular mechanisms still remain unclear, particularly for with regards to immunogenomic effects. This is the reason that we selected the IRGs.
We performed a multivariate cox regression to select out ten immune genes which are independent influencer to the survival to construct a prognostic model which is risk score=the plus of each gene expression multiply coefficient. The median of risk score of 212 cervical cancer is the cut off value of high risk group and low risk group.
We adopt Surv () function to do multivariate COX regression analysis performed by R. The R package of Surv (time, event) function is downloaded.
- Comment: The discussion repeats many elements from the introduction and should focus more on the identified IRGs in the analysis and their significance for predicting prognosis.
Response: According to your nice suggestions, we have made extensive corrections to our previous draft. We streamed the information about SIRGs. We supplemented the discussion about prognostic model and immunocytes infiltration. The detailed corrections are in yellow below.
Cervical cancer is caused by the persistent infection of hrHPV [15]. Although surgery, chemoradiotherapy, anti-angiogenic medicine and even the new immunotherapy have been applied for cervical cancer treatment, the prognosis remains poor at the late stage [4, 9, 16]. Therefore, research on effective prognostic biomarkers and new molecular mechanisms has drawn increasing attention. Kidd EA et al. have found that the standardized uptake value for F-18 fluorodeoxyglucose is a sensitive predictive biomarker for the survival of cervical cancer patients [17]. Luo W et al. have identified a 6 lncRNAs signature, which can be regarded as novel diagnostic biomarkers for cervical cancer [18]. Li X et al. have identified a histone family gene signature for predicting the prognosis of cervical cancer patients [19]. These studies provide an elemental knowledge of the pathogenesis of cervical cancer at the genetic level. However, the prognostic role of immunogenomics in cervical cancer remains largely undetermined. In the present study, we performed a comprehensive analysis of IRGs in cervical cancer, which might enhance our knowledge of their clinical value and help us understand potential molecular mechanisms. Moreover, these IRGs might act as valuable clinical biomarkers or therapeutic targets. Besides, we constructed a prognostic model that could help assess immune cell infiltration and potential clinical outcomes of cervical cancer patients.
Tumor immune microenvironment can promote the progression of cervical cancer, including cancer cell proliferation, invasion, metastasis, immunosuppression and tissue remodeling, fibrosis, and angiogenesis. For example, CXCL12 induces mononuclear phagocytes to release HB-EGF, triggering anti-apoptotic and proliferative signals in Hela cells [20]. D-dopachrome tautomerase (D-DT), a homolog of macrophage migration inhibitory factor (MIF), can promote the invasion of cervical cancer cells when it is overexpressed [21]. An altered balance in IL-12p70 and IL-10 production can weaken T cell proliferation in cervical cancer [22]. These studies suggest the importance of immunity in cervical cancer progression. Therefore, it is necessary to identify differentially expressed IRGs. Genome profile alterations cause tumorigenesis. We identified alternations in immunogenomic profiles to study the effect of alternations on the immune microenvironment and clinical prognosis. Gene functional enrichment analysis suggested that these genes were mainly involved in growth factor (GF) activity. GFs actively act in the pathogenesis of cervical cancer. Notably, these GFs are correlated to proliferation, aggression and migration [23,24,25]. Therefore, these GFs could also be used to monitor metastasis, assessing survival, and identify potential drug targets as clinical biomarkers.
Among the 20 SIRGs, no related reports have explored the roles of DUOX1, GRN, CSK, CD79, NFATC4, EPGN, TGFA, IL17RD, LEPR, N2RF1 and TRAV26-1 in cervical cancer. APOD is down-regulated in cervical cancer compared with normal cervix[26]. TFRC expression is up-regulated in cervical cancer compared with normal cervix[27]. A significant correlation between cervical cancer and the polymorphism of rs1041981 in the LTA gene has been observed[28]. The rare allele (A) of SNP rs2239704 in the 5′ UTR of the LTA gene is significantly associated with increased risks of cervical cancer[29]. F2RL1 is overexpressed in cervical cancer cell lines and significantly correlated with poor OS[30,31]. HGF overexpression in lesions of cervical cancer has been reported to be related to a poorer prognosis[32]. HDAC1/DNMT3A-containing complex is associated with the suppression of cancer stem cells in cervical cancer [33]. HGF can induce migration and invasion of cervical cancer cells [34].] BMP6 may participate in invasion and metastasis in cervical cancer [35]. The proportion of CD123(+) dendritic cells is significantly lower in the peripheral blood of cervical cancer patients compared with the controls [36]. A higher frequency of Nrp1(+) T-regs frequency suppress the immune response against distant cervical cancer cells [37]. The above-mentioned results are consistent with our current findings. However, Tyk2 is confirmed to be overexpressed in SCC[38], which was different from our study. Low-throughput experiments, such as Western blotting analysis, is required to verify the factual expression. PTPN6 is positively correlated with HPV infection in cervical cancer with the explanation of cell defense reaction [39].
To explore molecular mechanisms underlying the potential clinical importance, we constructed a TF-mediated network that could regulate hub IRGs. Among the SDETFs, Foxp3 significantly associated with FIGO stage and tumor size [40]. Foxp3 is associated with lymphangiogenesis of cervical cancer[41]. FoxP3 has been confirmed to be highly expressed in cervical cancer, and it facilitates the proliferation and invasiveness and inhibits the apoptosis of cervical cancer cells [42]. In conclusion, Foxp3 is a risk factor for the survival of cervical cancer, which is consistent with our current finding. CBX7 inhibits the proliferation of cervical cancer cells [43]. LTA inhibits the proliferation of CD4(+) T-cells in a FoxP3(+) Treg-dependent manner in patients with chronic hepatitis C, suggesting that LTA acts on FoxP3[44]. Therefore, previous studies provide limited information about the mechanisms of 10 IRGs in the survival of cervical cancer.
The effects of the JAK/STAT pathway and the persistent activation of STAT3 and STAT5 during the process of tumor cell proliferation, cycling and invasion have made it a favorite treatment target. In cervical cancer, the activated JAK/STAT signaling pathway by Bcl-2 promots cell viability, migration, and invasion [45]. There is a strong association between HPV infection and STAT-3 overexpression in cervical cancer [46]. The expression of STAT3 has been proposed as a poor prognostic factor in cervical cancer [47]. STAT5 protein is up-regulated and associated with the severity of cervical cancer [48]. Moreover, overexpression of STAT-5 elevates the STAT-3 expression compared with the normal controls [46]. Therefore, JAK/STAT signaling may play an important role in cervical carcinogenesis. Moreover, up-regulation of PAR2 (F2RL1) induces the proliferation of cervical cancer cells by activating STAT3 [49], which is consistent with our current results that overexpression of F2RL1 was involved in the JAK/STAT signaling pathway.
In the present study, we created an immune-based prognostic signature to monitor the immune status and access the prognosis for cervical cancer patients. Previously, Wu HY et al. (2020) have constructed a prognostic index based on percent-splice-in values in SCC [50]. Eun Jung Kwon et al. (2020) have explored genomic alterations and developed a risk index model can monitor HPV- related bladder cancer [51]. Cai LY has created a risk score model based on differentially expressed glycolysis-related genes, and the model can predict the prognosis of cervical cancer patients [52]. Recently, Zhao S et al. (2020) have constructed a 4-gene prognostic risk score model in CESC by identifying DEGs [53]. Beyond the above-mentioned studies, there are also many studies about prognostic model [54,55]. Compared with the previous studies, our prognostic model could also assess the immunocyte infiltration based on not only genomic alterations but also immune-genomic profiles. Moreover, we constructed a Tf-mediated regulatory network, which provided a more detailed mechanism of IIRGs. Our prognostic index, based on 10 differentially expressed IIRGs in cervical cancer, demonstrated favorable clinical viability. Our data showed that the risk score model performed moderately and steadily in prognostic predictions in patients with early-staged cervical cancer patients.
Among these 10 IIRGs, five IIRGs were significantly correlated with immunocyte infiltration, especially neutrophils, dendritic cells, CD4+ T cells, macrophages, and B cells, indicating that a greater immunocyte infiltration might be observed in low-risk patients. Our results confirmed immunocyte infiltration promoted tumor progression. Many studies have investigated the role of immunocytes in cervical cancer. Zhou PJ et al. (2020) have pointed out a high neutrophil to- lymphocyte ratio is independently associated with decreased OS and PFS in patients with cervical cancer by meta-analysis [56]. Ma Y et al. (2013) have shown that dendritic cells exhibit tolerogenic or immunosuppressive states that favor malignant progression[57]. K Manjgaladze et al. (2019) have proved that the infiltration of both CD4 T helper cells and CD8 cytotoxic T cells is significantly increased during the progression of cervical intraepithelial neoplasia [58]. M2-like macrophages are increasingly infiltrated in cervical cancer [59]. O'Brien PM et al. (2001) have shown that B-lymphocytes are the predominant lymphocyte infiltrate in pre-malignant cervical lesions to compromise the host immune response [60]. These results are consistent with our conclusions. However, the role of immune cells in cervical cancer still remains largely unclear. Further research is needed.
We must point out that only three control specimens were acquired in the present study. Although it met the minimum requirements for biological repeat, insufficient control samples tended to cause larger errors. Therefore, more other experiments are still necessary to validate the transcriptome results.
We tried our best to improve the manuscript and made some changes in the manuscript. These changes will not influence the content and framework of the paper. And here we did not list the changes but marked in yellow in revised paper. We appreciate for Editors/reviewers’ warm work earnestly, and hope that the correction will meet with approval. Once again, thank you very much for your comments and suggestions.

Round 2
Reviewer 1 Report
Abstract:
I do not see the logic between sentence two and three. Do you mean that it is important to find predictive markers within the tumor immune microenvironment? If so, please state so in sentence 3. Suggestion: Therefore, it is necessary to comprehensively identify predictive biomarkers associated with cervical cancer prognosis, also within the immune microenvironment.
Introduction:
Line 53: You should include ‘most’ before ‘frequently’.
Line 65: Typo. Please change to enhanced if that is what you mean.
Discussion:
Thank you for the info on the para-tumor samples and for the added paragraph on study limitations related to the small sample size (n=3). However, I still cannot see that you have commented on the tumor cell content in the para-tumor samples. It should be 0% or preferably not more that 5%. Also, it would be interesting to know more about the different cell types in the para-tumor samples. By that I mean the percentage of stromal cells, squamous/glandular epithelium and immune cells. If this in not available, I would recommend including this as one of the study limitations.
Figures: The layout of the figures should be improved.
Reviewer 2 Report
Thank you for your efforts to revise your manuscript as reviewers' comments.